# Interleukin-10 contributes to PGE$_2$ signalling through upregulation of EP4 via SHIP1 and STAT3

Abrar Samiea[1,2°], Jeff S. J. Yoon[1,2,3°], Sylvia T. Cheung[1,2,3], Thomas C. Chamberlain[1,2,3‡], Alice L. -F. Mui[1,2,3‡]*

1 Immunity and Infection Research Centre, Vancouver Coastal Health Research Institute, Vancouver, Canada, 2 Department of Surgery, University of British Columbia, Vancouver, Canada, 3 Department of Biochemistry and Molecular Biology, University of British Columbia, Vancouver, Canada

° These authors contributed equally to this work.
‡ These authors are joint senior authors on this work.
* alice.mui@ubc.ca

**Data Availability Statement:** All relevant data are within the manuscript.

**Funding:** These studies were supported by operating grants from the Canadian Institutes of

## Abstract

Macrophage cells form part of our first line defense against pathogens. Macrophages become activated by microbial products such as lipopolysaccharide (LPS) to produce inflammatory mediators, such as TNFα and other cytokines, which orchestrate the host defense against the pathogen. Once the pathogen has been eradicated, the activated macrophage must be appropriately deactivated or inflammatory diseases result. Interleukin-10 (IL10) is a key anti-inflammatory cytokine which deactivates the activated macrophage. The IL10 receptor (IL10R) signals through the Jak1/Tyk2 tyrosine kinases, STAT3 transcription factor and the SHIP1 inositol phosphatase. However, IL10 has also been described to induce the activation of the cyclic adenosine monophosphate (cAMP) regulated protein kinase A (PKA). We now report that IL10R signalling leads to STAT3/SHIP1 dependent expression of the EP4 receptor for prostaglandin E$_2$ (PGE$_2$). In macrophages, EP4 is a G$_{αs}$-protein coupled receptor that stimulates adenylate cyclase (AC) production of cAMP, leading to downstream activation of protein kinase A (PKA) and phosphorylation of the CREB transcription factor. IL10 induction of phospho-CREB and inhibition of LPS-induced phosphorylation of p85 PI3K and p70 S6 kinase required the presence of EP4. These data suggest that IL10R activation of STAT3/SHIP1 enhances EP4 expression, and that it is EP4 which activates cAMP-dependent signalling. The coordination between IL10R and EP4 signalling also provides an explanation for why cAMP elevating agents synergize with IL10 to elicit anti-inflammatory responses.

## Introduction

Macrophages participate in host defense [1–4] by providing an immediate response to invading pathogens while also contributing to the activation of the adaptive immune system [5–7]. When lipopolysaccharide (LPS), a component of the outer membrane of gram negative

Health Research (MOP-133415) and the Natural Science and Engineering Research Council (RGPIN-2014-05662). The funders had no role in study design, data collection and analysis, decision to publish, or preparation of the manuscript.

**Competing interests:** The authors have declared that no competing interests exist.

bacteria, binds to the Toll-like receptor 4 (TLR4) on the macrophage, a cascade of signalling pathways is initiated leading to the production of pro-inflammatory cytokines and other inflammatory mediators [8]. However, this inflammatory response needs to be appropriately terminated or inflammatory disorders can develop [9]. The anti-inflammatory cytokine interleukin-10 (IL10) is a key suppressor of immune cell activation vital to maintaining immune homeostasis [9, 10]. Defects in IL10 signalling are associated with various autoimmune/inflammatory diseases [11–13]. IL10 knockout mice develop colitis, comparable to human inflammatory bowel disease [14, 15] and are hypersensitive to inflammatory stimuli [16]. Deficiencies or mutations in IL10 [17] or the IL10 receptor (IL10R) [12, 18] are found to cause early onset of inflammatory bowel disease (IBD) in humans.

IL10 initiates signalling by binding to its receptor (IL10R), activating receptor associated Jak1 and Tyk2 tyrosine kinases [19, 20], and leading to activation of the STAT3 transcription factor [21–23]. We have shown that in addition to the STAT3 pathway, IL10R signalling also involves the SH2 domain containing inositol 5' phosphatase 1 (SHIP1) [24, 25]. SHIP1 is a cytoplasmic protein expressed predominantly in hematopoietic cells where one of its functions is to negatively regulate phosphoinositide 3-kinase (P13K) signalling by hydrolyzing the PI3K product phosphatidylinositol-3,4,5-trisphosphate (PIP3) into PI-3,4-bisphosphate (PI-3,4-P$_2$) [26–28]. SHIP1 can also serve as an adaptor protein for assembly of protein complexes in signalling pathways [29]. We found that IL10 signalling requires SHIP1 to inhibit LPS-initiated translation of pre-existing TNFα mRNA [24] and the maturation of the microRNA, miR155 [25]. Recently, we showed that SHIP1 and STAT3 form complexes following IL10 stimulation to mediate IL10 responses and that small molecule allosteric activators of SHIP1 can mimic the anti-inflammatory actions of IL10 *in vitro* and *in vivo* (Chamberlain *et al.*, submitted).

In studies to further characterize the IL10-regulated gene products which participate in the anti-inflammatory response, we used DNA microarrays to screen for mRNAs whose regulation by IL10 involved SHIP1. We found the mRNA for *ptger4* was upregulated by IL10 ~2.5 fold more in SHIP1 $^{+/+}$ than $^{-/-}$ mouse peritoneal macrophages (perimacs). *Ptger4* encodes the 513 amino acid, prostaglandin E$_2$ (PGE$_2$) receptor 4 (EP4), a G protein-coupled receptor, reported to have anti-inflammatory actions [30–32]. EP4 is one of four surface prostaglandin (PG) E$_2$ receptors of the sub-group, E-type prostaglandin receptors (EP1-4) [33]. EP4, but not the other PGE$_2$ receptors, is abundantly expressed in macrophages, suggesting a role in regulating immune responses [30, 34]. EP4 in macrophages is coupled to the G$_{αs}$ trimeric G protein. Binding of PGE$_2$ to EP4 results in activation of adenylate cyclase (AC), elevation of cAMP levels, activation of cAMP-dependent protein kinase A (PKA) and phosphorylation of the CREB transcription factor [33].

Studies of siRNA mediated EP4 knock-down cells [35] and EP4$^{-/-}$ mice [31, 36] suggest that EP4 is responsible for the anti-inflammatory activity of PGE$_2$. EP4$^{-/-}$ mice are more susceptible to induced experimental colitis [31]. EP4 but not EP1-3 mRNA levels rise in colitic mice [31, 32] suggesting an important role for EP4 but not the other PGE$_2$ receptors in colitis. In humans, genome-wide association studies (GWAS) have correlated EP4 polymorphisms with inflammatory diseases including Crohn's disease [37–39] and allergic diseases [40]. EP4-specifc agonists inhibited colitis in wild type mice [31] and EP4 agonists have been developed for treating inflammatory disease in humans, with Rivenprost (ONO-4819CD, AE1-734) showing efficacy in phase II trials in ulcerative colitis [41].

These observations suggest a protective role of EP4 that is similar to IL10-mediated anti-inflammatory actions. We now present evidence IL10-induced upregulation of EP4, and subsequent EP4 signalling plays a role in the overall anti-inflammatory actions of IL10 in macrophages. IL10 utilizes both STAT3 and SHIP1 to upregulate EP4 protein and knock-down of EP4 is required for mediating some of the anti-inflammatory actions of IL10.

## Materials and methods

### Mouse colonies

BALB/c mice of SHIP1 wild type ($^{+/+}$) or SHIP1 knockout ($^{-/-}$) were kindly provided by Dr. Gerald Krystal (BC Cancer Research Centre, Vancouver, BC). C57BL/6 STAT3$^{-/-}$ mice were generated by crossing C57BL/6 STAT3$^{flox/flox}$ mice (Dr. Shizuo Akira, Hyogo College of Medicine, Nishinomiya, Japan) with C57BL/6 LysMCre mice (Jackson Laboratory). Offspring of these mice were heterozygous on both alleles, and were then crossed with homozygous STAT3-$^{flox/flox}$ mice to generate mice with a genotype of STAT3$^{flox/flox}$/LysMCre$^{+/-}$. Then, STAT3$^{flox/}$ $^{flox}$/LysMCre$^{+/-}$ mice were crossed with STAT3$^{flox/flox}$ mice to generate both STAT3$^{flox/flox}$ /LysMCre$^{+/-}$ mice (STAT3$^{-/-}$ mice) and STAT3$^{flox/flox}$ mice (STAT3$^{+/+}$ mice) in the same litters. For mRNA analysis and immunoblotting studies, BALB/c mice $^{+/+}$ or $^{-/-}$ for SHIP1 and C57BL/ 6 LysMCre mice $^{+/+}$ or $^{-/-}$ for STAT3 were used. Both male and female were used, aged between 6–20 weeks old. All mice were housed and maintained in accordance with the ethic protocols approved by the University of British Columbia Animal Care Committee.

### Cells

RAW264.7 cells were obtained from the American Type Culture Collection and maintained in Roswell Park Memorial Institute medium (RPMI-1640) (HyClone, Logan, Utah) supplemented with 9% fetal bovine serum (FBS) (HyClone, Logan, Utah). Primary peritoneal macrophages (perimacs) were isolated from mice by peritoneal lavage with 3 ml of sterile Phosphate Buffered Saline (PBS) (HyClone, Logan, Utah). Perimacs were collected and transferred to Iscove's Modified Dulbecco's Medium (IMDM) (HyClone, Logan, Utah) supplemented with 10% (v/v) FBS, 10 μM β-mercaptoethanol, 150 μM monothioglycolate and 1 mM L-glutamine (referred to here on as Mac media). Bone marrow-derived macrophages (BMDM) were generated by first collecting femurs and tibias from mice, and then flushing out the bone marrow through a 26-G needle. Extracted cells were plated, in Mac media supplemented with 5 ng/ml each of CSF-1 and GM-CSF (Stem Cell Technologies, Vancouver, BC), on a 10-cm tissue culture plate for 2 hours at 37˚C. Non-adherent cells were collected and replated at 9×10$^6$ cells per 10-cm tissue culture plate. Cells were then cultured in the presence of CSF-1 and GM-CSF. Differentiated BMDMs were used after 7 to 8 days. All cells were maintained in a 37˚C, 5% CO2, 95% humidity incubator.

### Constructs

Lentiviral expression vectors for the doxycycline (Dox) inducible CRISPR/Cas9 and sgRNA were purchased from Addgene (Lenti-iCas9-neo #85400; pLX-sgRNA #50662 [42, 43]). Guide sequences used in the present study to target EP4 gene were designed via CRISPR Gold online tool [44]. Two guide RNA sequences were designed to target different positions of EP4 gene: EP4 KD1 sgRNA (GGCGGCGTAGGCCGTTACGT), EP4 KD2 sgRNA (CGACTTGCACAATAC TACGA). Target sequences were cloned into the pLX-sgRNA vector using overlap-extension PCR to generate sgRNA-specific inserts. In brief, the first PCR amplicons were produced from F1/R1 and F2/R2 primers using Phusion polymerase (ThermoFisher Scientific, Nepean, ON). The first PCR products were then gel purified and extracted using phenol:chloroform:isoamyl (25:24:1) alcohol (ThermoFisher Scientific, Nepean, ON) to remove any template vectors and Phusion polymerases that can interfere with producing correct clones in proceeding steps. The first PCR products were then used as templates for another PCR reaction with the F1/R2 primer pair and the same gel purification and phenol-chloroform extraction was performed. The product of second PCR reaction was digested using NheI and XhoI restriction enzymes,

ligated into the empty pLX-sgRNA vector and transformed into chemically competent Stbl3 bacteria. Ampicillin resistant colonies were selected and the sequences confirmed by sequencing. The virus containing Lenti-iCas9 neo plasmid was prepared by mixing the packaging plasmid R8.9 and VSVG with Lenti-iCas9 neo plasmid and transfecting into HEK293T cells to produce virus [25]. The lentivirus in the media was collected and concentrated to use to infect RAW264.7 cells in the presence of 8 μg/ml protamine sulfate. The Lenti-iCas9 neo transduced RAW264.7 cells were selected by sorting for GFP fluorescence using FACS. After sorting, cultures were maintained in 2 mg/ml neomycin. Then, pLX-sgRNA vectors with EP4 target sequences were infected into the RAW264.7 cells containing Lenti-iCas9 neo plasmids using the same procedure. The RAW264.7 cells with pLX-sgRNA vector were selected using 5 μg/ml blasticidin. To induce the expression of Cas9, 2 μg/ml doxycycline (Dox) was added to the culture media for 48 hours. All cell lines were maintained at 37°C, 5% CO2 and 95% humidity.

## Reagents

Antibodies used include SHIP1 (P1C1) mouse antibody (Santa Cruz Biotechnology, Santa Barbara, CA), pSTAT3 Y705 (3E2) mouse antibody (Cell Signaling, Danvers, MA), STAT3 (79D7) rabbit antibody (Cell Signaling, Danvers, MA), EP4 (Santa Cruz Biotechnology, Santa Barbara, CA), p-p85 PI3 Kinase (Tyr458) rabbit antibody (Cell Signaling, Danvers, MA), p85 PI3 Kinase rabbit antibody (Cell Signaling, Danvers, MA), p-p70 S6 Kinase (Thr389) rabbit antibody (Cell Signaling, Danvers, MA), p70 S6 Kinase rabbit antibody (Cell Signaling, Danvers, MA), pCREB (Ser133) rabbit antibody (Cell Signaling, Danvers, MA), CREB (48H2) rabbit antibody (Cell Signaling, Danvers, MA), and GAPDH rabbit antibody (Sigma, Oakville, ON)..

## Cell stimulations for immunoblot analysis

RAW264.7 cells and BMDMs were seeded at 3.0 x 10$^5$ cells per well on 24-well tissue culture plates 1 day prior to stimulation, followed by replacement with fresh medium for 1 hour prior to stimulation with 10 ng/ml LPS (*Escherichia coli* serotype 0111:B4; Sigma) with and without the indicated concentrations of IL10 for the required time points. Perimacs were seeded at 1.0 x 10$^6$ cells per well in 24-well tissue culture plate and let to adhere for 3 hours before washing with PBS to remove non-adherent cells followed by stimulation with 10 ng/ml LPS +/- 10 ng/ml IL10 in Mac media for 1 hour. The EP4 Plx-sgRNA transduced cell lines were left untreated or treated with 2 μg/ml Dox for 24 hours prior to cell seeding (a total of 48 hours treatment before LPS/IL10 stimulations) to induce knockdown of EP4.

## Treatment with EP4 antagonist

Cells were seeded at 2.0 x 10$^4$ cells per well in a 96-well tissue culture plate and allowed to adhere overnight. Media was changed the next day 1 hour prior to stimulation. For cells that need pre-treatment, DMSO (final concentration of 0.05%) or 100 nM ONO-AE3-208 (14522, Cayman Chemical Company, Ann Arbor, MI) prepared in RPMI-9% FBS media was used to pre-treat the cells for 1 hour prior to stimulations. Cells were then stimulated with 1 ng/ml LPS +/- IL10 (0.01–30 ng/ml) for 1 hour. Triplicates wells were used for each stimulation condition. Supernatant was collected and secreted TNFα protein levels were measured by ELISA as described below.

## Real-time quantitative PCR

Perimacs were seeded at 3.0 x 10$^6$ cells/well in 6-well tissue culture plates and allowed to adhere for 3 hours before washing with PBS to remove non-adherent cells followed by stimulation

with 1 ng/ml LPS +/- 10 ng/ml IL10 in Mac media for 1 hour. BMDM and RAW264.7 cells were seeded at 2 x 10$^6$ cells/well and 1.5 x 10$^6$ cells/well respectively. Total RNA was extracted using Trizol reagent (ThermoFisher, Nepean, ON) according to manufacturer's instructions. 3 µg of RNA were treated with DNAse I (Roche Diagnostics, Laval, QC) according to the product manual. For mRNA expression analysis, 120 ng of RNA were used in the Transcriptor First Strand cDNA synthesis kit (Roche Diagnostics, Laval, QC), and 0.1 µl to 0.2 µl of cDNA generated were analyzed by SYBR Green-based real time PCR (real time-PCR) (Roche Diagnostics, Laval, QC) using 300 nM of gene-specific primers—EP4 Forward (`ACCATTCCTAGA TCGAACCGT`), EP4 Reverse (`CACCACCCCGAAGATGAACAT`), GAPDH Forward (`AATG TGTCCGTCGTGGATCT`), GAPDH Reverse (`GCTTCACCACCTTCTTGATGT`). Expression levels of mRNA were measured with the StepOne Plus RT-PCR system (Applied Biosystems, Burlington, ON), and the comparative Ct method was used to quantify mRNA levels using GAPDH as the normalization control.

## Immunoblot analysis

Cells were rinsed with cold PBS and lysed with hot 2 x Laemmli sample buffer. Proteins were separated by 12.5% SDS-PAGE, followed by electroblotting onto polyvinylidene fluoride (PVDF) membrane (Millipore, Etobicoke, ON). Membranes were blocked with 3% BSA/TrisHCl buffer saline (TBS)/pH 7.5 (blocking buffer), rinsed with 0.05% Tween 20/TBS (wash buffer) and probed primary antibodies/3% blocking buffer at room temperature overnight. The following day, membranes were washed 3 X 10 min, incubated 1 hour at room temperature with Alexa Fluor® 660 anti-mouse IgG or Alexa Fluor® 680 anti-rabbit IgG antibodies in wash buffer (ThermoFisher, Nepean, ON), washed and imaged using a LI-COR Odyssey Imager.

## Measurement of TNFα production and IL10 IC50 calculations

Cells were seeded at 2.0 x 10$^4$ cells per well in a 96-well tissue culture plate and allowed to adhere overnight. Media was changed the next day 1 hour prior to stimulation. Cells were stimulated with 1 ng/ml LPS +/- IL10 (0.01–30 ng/ml) for 1 hour. Triplicates wells were used for each stimulation condition. Supernatant was collected and secreted TNFα protein levels were measured using a BD OptEIA Mouse TNFα Enzyme-Linked Immunosorbent Assay (ELISA) kit (BD Biosciences, Mississauga, ON). The inhibitory concentration (IC50) of IL10 needed to inhibit LPS-stimulated TNFα production was calculated using GraphPad (non-linear regression curve fit). The mean IC50 was determined from three independent IC50 determination experiments.

## Statistical analysis

Quantification of band intensities in immunoblots was performed using LI-COR Odyssey imaging system and Image Studio™ Lite software (LI-COR Biosciences, Lincoln, NE). GraphPad Prism 6 (GraphPad Software Inc., La Jolla, CA) was used to perform all statistical analyses. Statistical details can be found in figure legends. Values are presented as means ± standard deviations. Unpaired $t$ tests were used where appropriate to generate two-tailed P values. Two-way ANOVA were performed where required with appropriate multiple comparisons tests. Differences were considered significant when $p \leq 0.05$.

## Ethics statement

Some of the studies required cells harvested from mice. The mice were kept, and cells were harvested in compliance with the Canadian Animal Care Committee guidelines.

## Results

### EP4 mRNA is upregulated by IL10 in a SHIP1 and STAT3 dependent manner

In studies to further characterize the IL10-regulated gene products which participate in the anti-inflammatory response, we looked at mRNA expression induced by IL10 in SHIP1$^{+/+}$ vs SHIP1$^{-/-}$ peritoneal macrophages (perimacs). We found that 341 genes were upregulated by IL10 by at least two-fold in SHIP1$^{+/+}$ macrophages. Of these, 124 were differentially regulated in SHIP1$^{-/-}$ cells by at least a two-fold difference. One IL10-regulated gene is the EP4 mRNA (encoded by the *ptger4* gene) whose expression was upregulated by IL10 ~2.5 fold more in SHIP1$^{+/+}$ than SHIP1$^{-/-}$.

To confirm our microarray finding that IL10 upregulates expression of EP4 mRNA, we performed qPCR analysis on RNA isolated from LPS +/- IL10 stimulated SHIP1 and STAT3 wild-type and knock-out perimacs. As shown in Fig 1A, at 1 hour LPS + IL10 upregulates EP4 gene expression in SHIP1$^{+/+}$ and STAT3$^{+/+}$ cells by 2.6 and 2.9 fold, respectively, compared to LPS alone. In contrast, IL10-induction of EP4 was impaired in both SHIP1$^{-/-}$ and STAT3$^{-/-}$ cells, by approximately 45% in SHIP1$^{-/-}$ cells (Fig 1A, left) and 30% in STAT3$^{-/-}$ cells (Fig 1A, right). This suggests that IL10-mediated expression of EP4 mRNA requires both SHIP1 and STAT3 for maximal induction. LPS by itself stimulated a slight increase (~ 0.5 fold more than the unstimulated condition) in EP4 mRNA.

Our biochemical studies will make use of the RAW264.7 cell line and bone marrow derived macrophages (BMDM) from SHIP1$^{+/+}$ or $^{-/-}$ and STAT3$^{+/+}$ or $^{-/-}$ mice, so we also examined EP4 mRNA levels in those cells. Similar to that observed in perimacs, IL10 +/- LPS (but not LPS alone) induction of EP4 mRNA is impaired in SHIP1$^{-/-}$ BMDM (Fig 1B, left). However, unlike that observed in SHIP1$^{-/-}$ perimacs, STAT3 deficiency in BMDM did not affect IL10's ability to elevate EP4 mRNA (Fig 1B, right). We confirmed that these STAT3$^{-/-}$ are indeed impaired in IL10 inhibition of TNFα mRNA (Fig 1C) as we and others have previously described ([25, 45]. Finally, we examined the responses in the RAW264.7 cell line. As seen in Fig 1D, the pattern of EP4 mRNA expression patterns similar to SHIP1$^{+/+}$ and STAT$^{+/+}$ perimacs and BMDM.

### IL10 dose-dependent upregulation of EP4 protein

Next, we examined whether IL10 upregulation of EP4 mRNA reflected changes in EP4 protein levels in the RAW264.7 macrophage cell line (Fig 2) Cells were treated with the indicated concentrations of IL10 for 1 hour in the presence of 10 ng/ml LPS. IL10 induced EP4 protein in a dose-dependent manner (Fig 2). The lowest concentration of IL10 tested (10 ng/ml) was found to produce significant upregulation of EP4, compared to cells with no IL10 with no further increases with higher IL10 concentrations. Therefore, this concentration was chosen for use in further experiments. The IL10 concentration dependence of EP4 induction paralleled the phosphorylation of STAT3 tyrosine 705 (pSTAT3-Y705). No EP4 protein was observed with LPS stimulation alone.

### Kinetics of EP4 upregulation

After determining a suitable concentration of IL10 to use to observe upregulation of EP4 protein levels, we tested different time points (from 0.5 to 2 hours) to understand the kinetics of IL10 induction of EP4. IL10-induced expression of EP4 was detected as early as 0.5 hour in RAW264.7 (Fig 3). EP4 induction peaked at 1 hour and started to decline afterwards (Fig 3). The kinetics of EP4 appearance mirrored the kinetics of pSTAT3-Y705 levels. Notably

**(A) Perimacs**

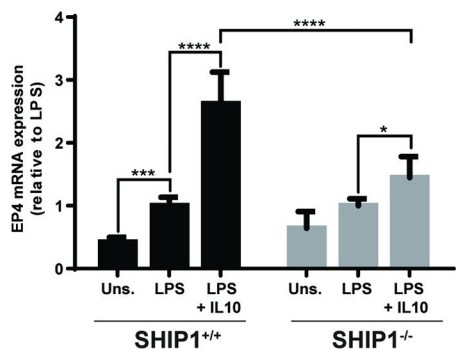

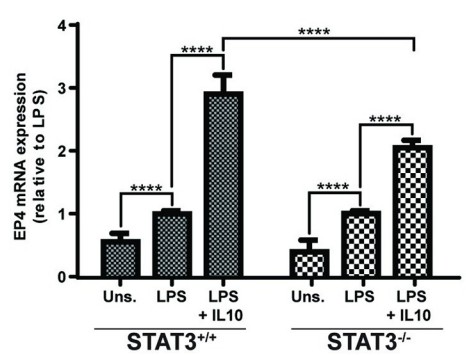

**(B) BMDM**

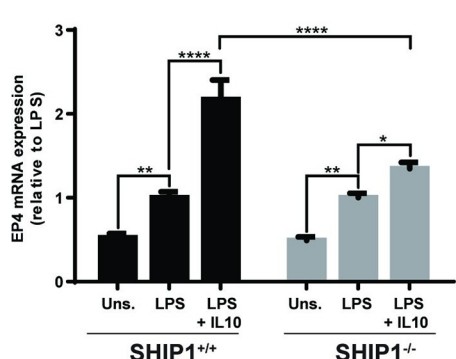

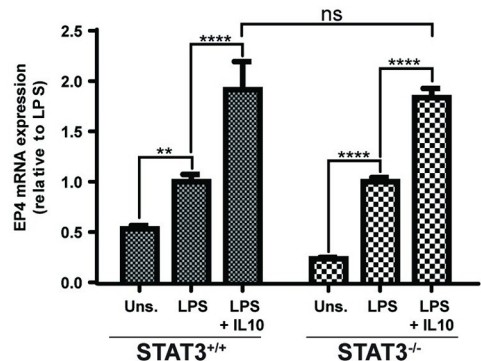

**(C) BMDM**

**(D) RAW264.7**

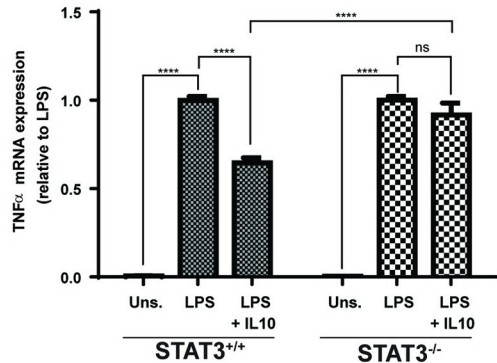

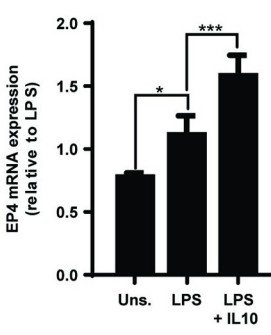

**Fig 1. SHIP1 and STAT3 dependence of IL10 induced expression of EP4 mRNA.** Perimacs extracted (A) or BMDM (B, C) derived from SHIP1$^{+/+}$ or $^{-/-}$ mice and STAT3$^{+/+}$ or $^{-/-}$ mice were stimulated with LPS +/- IL10 for 1 hour prior to total RNA extraction. Expression level of EP4 (A, B) or TNFα (C) mRNA was determined by real-time PCR and normalized to GAPDH mRNA levels. (Two-Way ANOVA with Tukey's correction, **** $p < 0.0001$, *** $p < 0.001$, ** $p < 0.01$, * $p < 0.05$, ns = not significant). RAW264.7 cells (D) were stimulated and subjected to the same mRNA expression analyses. (One-Way ANOVA with Tukey's correction, **** $p < 0.0001$, *** $p < 0.001$, ** $p < 0.01$, * $p < 0.05$, ns = not significant). Data represents expression levels relative to LPS stimulated cells from three independent experiments in each cell type.

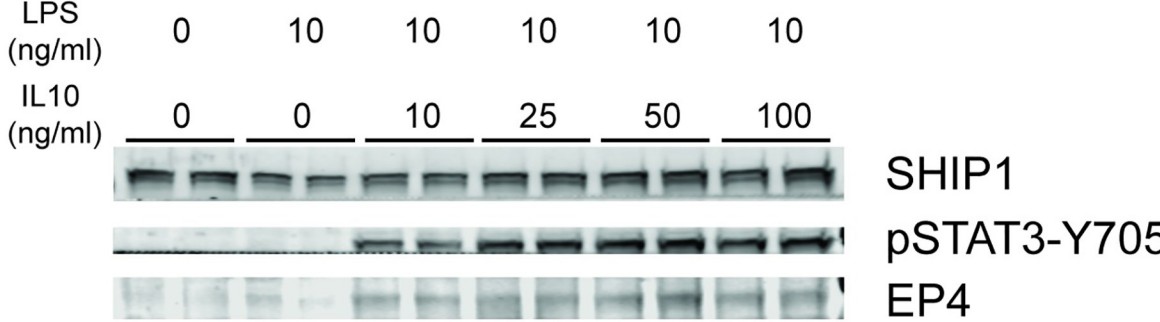

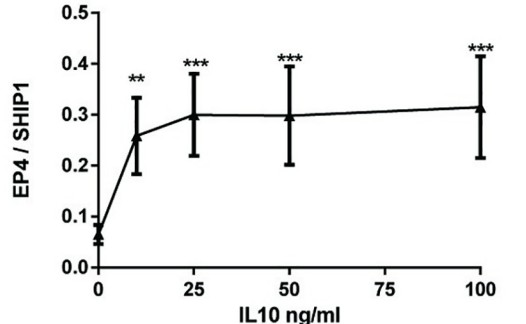

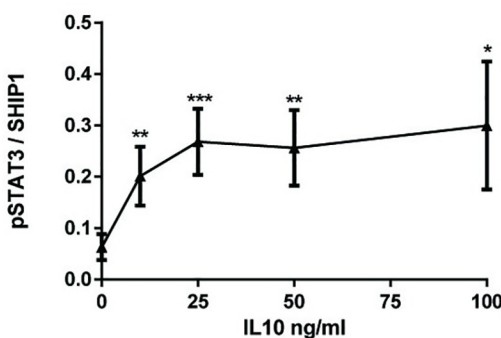

**Fig 2. IL10 dose-dependent upregulation of EP4 protein.** RAW264.7 cells were stimulated with 10 ng/ml LPS with the indicated concentrations of IL10 for 1 hour prior to protein lysate collection. Expression levels of EP4 and pSTAT3 proteins were determined by immunoblotting. Data plotted represents EP4 and pSTAT3 band intensities normalized to SHIP1 protein levels. The significance of the comparison between LPS and LPS + IL10 treatments was calculated by Two-Way ANOVA with Tukey's correction, **** $p<0.0001$, *** $p<0.001$, ** $p<0.01$, * $p<0.05$, ns = not significant. Data are representative of three independent experiments.

although we observed a low level of EP4 RNA upregulation in LPS treated SHIP1$^{+/+}$ and STAT3$^{+/+}$ cells ([Fig 1]), EP4 protein is detected only in the LPS + IL10 samples suggesting that IL10 is required for stable expression of EP4 protein.

## IL10 upregulation of EP4 protein requires SHIP1 and STAT3

We found that EP4 mRNA was modestly induced by LPS and strongly upregulated by IL10 in SHIP1$^{+/+}$ and STAT3$^{+/+}$ perimacs but less so in cells deficient of either SHIP1 or STAT3 ([Fig 1]). We next examined whether IL10-mediated upregulation of EP4 protein was also impaired in SHIP1$^{-/-}$ or STAT3$^{-/-}$ cells. IL10 increased EP4 levels in both SHIP1$^{+/+}$ perimacs ([Fig 4A]) and BMDM ([Fig 4B]) after 1 hour treatment; SHIP1 deficiency impaired induction in both cell types. No EP4 protein was induced by LPS alone in perimacs ([Fig 4A]). In BMDM we saw a basal expression of EP4 protein. This might be due to low level constitutive IL10 production in BMDM which prompt some investigators to use IL10$^{-/-}$ cells in their studies [23, 46], but other factors may also be involved. Regardless, the addition of IL10 increased the amount of EP4 protein in both SHIP1$^{+/+}$ and STAT3$^{+/+}$ cells. IL10-mediated EP4 induction is impaired in both SHIP1$^{-/-}$ and STAT3$^{-/-}$ BMDMs ([Fig 4C]). The impaired IL10 induction of EP4 *protein* in STAT3$^{-/-}$ BMDMs contrasts with the lack of effect of STAT3 deficiency on IL10 induction of EP4 *mRNA* in STAT3$^{-/-}$ perimacs. We also found that the level of STAT3 phosphorylation was

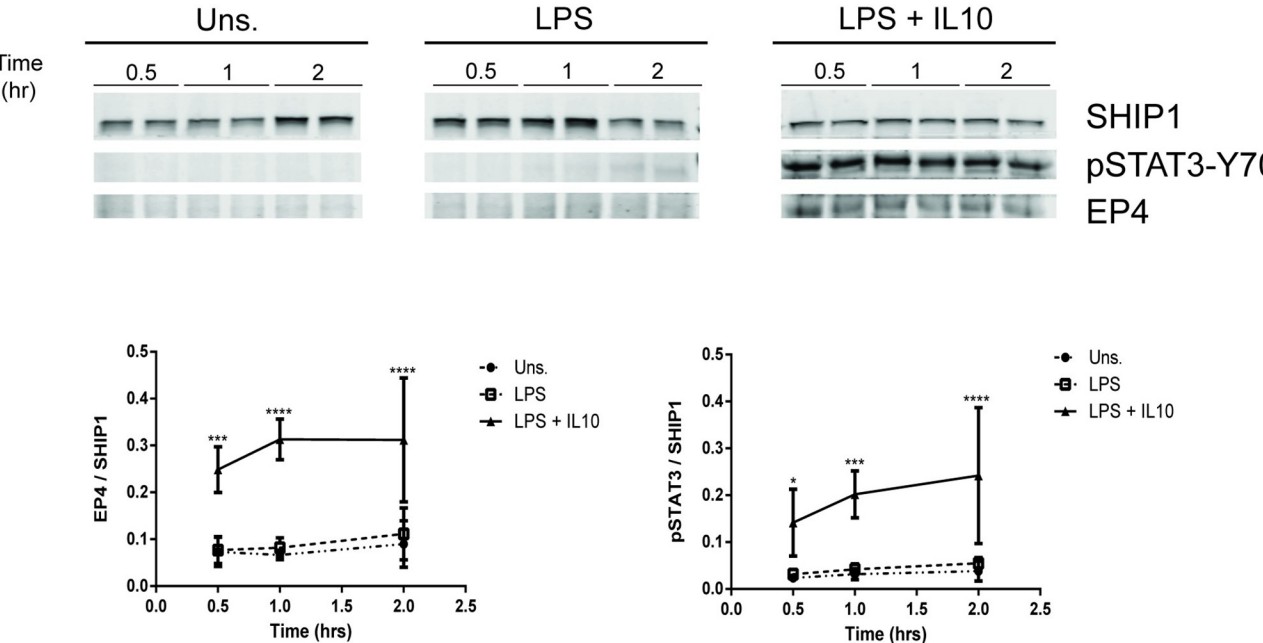

**Fig 3. Kinetics of EP4 upregulation.** RAW264.7 cells were stimulated with 10 ng/ml LPS ± 10 ng/ml IL10, at the indicated time points, prior to protein lysate collection. Expression levels of EP4 and pSTAT3 proteins were determined by immunoblotting. Data plotted represents EP4 and pSTAT3 band intensities normalized to SHIP1 protein levels. The significance of the comparison between LPS and LPS + IL10 treatments was calculated by Two-Way ANOVA with Tukey's correction, **** $p < 0.0001$, *** $p < 0.001$, ** $p < 0.01$, * $p < 0.05$. Data are representative of three independent experiments.

significantly lower in perimac and BMDM cells deficient in SHIP1, suggesting a role for SHIP1 in the phosphorylation of STAT3 in response to IL10.

## EP4 is required to mediate IL10 action in LPS-activated macrophages

To investigate whether EP4 contributes to IL10's anti-inflammatory action, we first made use of the EP4 antagonist ONO-AE3-208 [47]. RAW264.7 cells were treated with 100 nM ONO-AE3-208 or with DMSO for 1 hour prior to stimulation with 1 ng/ml LPS with 0–30 ng/ml IL10 for 1 hour. Culture supernatants were collected and the TNFα levels in them determined by ELISA. As shown in Fig 5, the IL10 inhibited LPS-stimulated TNFα expression in the control and DMSO treated cells. The ability of IL10 to inhibit TNFα expression was significantly impaired at 0.1 and 1 ng/ml of IL10 (Fig 5). Of note, ONO-AE3-208 had no effect at 30 ng/ml of IL10. 30 ng/ml is the concentration at which maximal EP4 protein levels is induced by IL10 (Fig 2).

Next we investigated the effect of EP4 knockdown on the phosphorylation of CREB which is a transcription factor downstream of PGE$_2$ stimulation of EP4-dependent elevation of cAMP levels in cells [33] and also implicated in IL10R signaling [48]. We found that LPS slightly, but LPS + IL10 strongly induced phosphorylation of CREB in no Dox cells (Fig 6A

To complement the chemical inhibitor studies, we also used CRISPR-Cas9 to knockdown EP4 in RAW264.7 cells. We generated RAW264.7 cells expressing a doxycycline inducible Cas9 protein (RAW264.7-Cas9) into which we introduced gRNAs KD1 and KD2 targeting two different exons of EP4 to generate the EP4-KD1 and EP4-KD2 cell lines. EP4 protein expression was knocked down by induction of Cas9 protein with doxycycline (+ Dox) for 48 hours (Fig 6A and 6B). We found that knocking down EP4 had little impact on IL10 phosphorylation of STAT3.

**(A)  SHIP1 $^{+/+}$ and $^{-/-}$ perimacs**

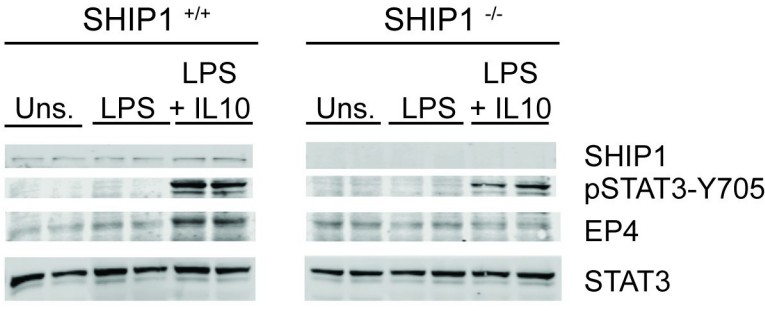
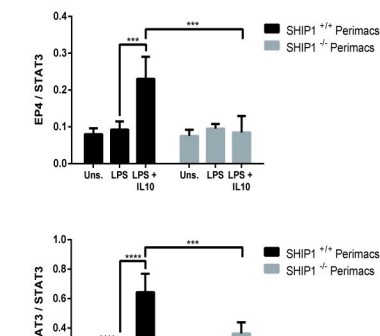

**(B)  SHIP1 $^{+/+}$ and $^{-/-}$ BMDMs**

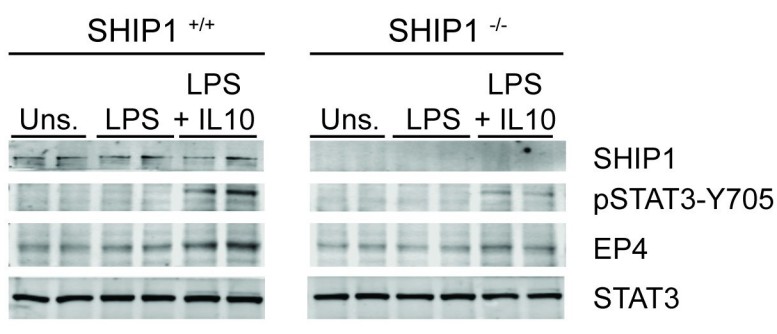
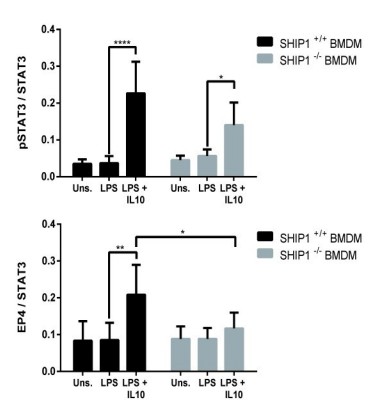

**(C)  STAT3 $^{+/+}$ and $^{-/-}$ BMDMs**

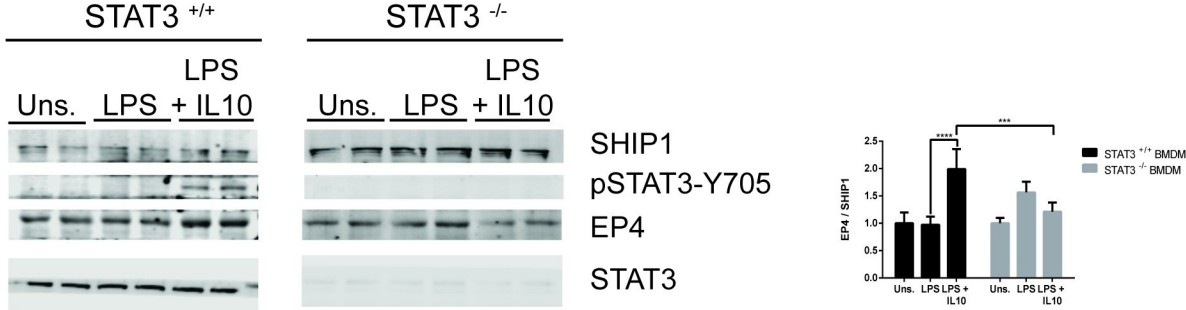

**Fig 4. IL10 upregulation of EP4 protein requires SHIP1 and STAT3.** (A) SHIP1$^{+/+}$ and $^{-/-}$ perimacs, (B) SHIP1$^{+/+}$ and $^{-/-}$ BMDM and (C) STAT3$^{+/+}$ and $^{-/-}$ BMDM were stimulated with 10 ng/ml LPS ± 10 ng/ml IL10 for 1 hour prior to protein lysate collection. Expression levels of EP4 and pSTAT3 proteins were determined by immunoblotting. Data plotted represents EP4 and pSTAT3 band intensities normalized to either STAT3 protein levels for (A) SHIP1$^{+/+}$ and $^{-/-}$ perimacs and (B) SHIP1$^{+/+}$ and $^{-/-}$ BMDM, or SHIP1 protein levels for (C) STAT3$^{+/+}$ and $^{-/-}$ BMDM. (Two-Way ANOVA with Tukey's correction, **** $p<0.0001$, *** $p<0.001$, ** $p<0.01$, * $p<0.05$). Data are representative of three independent experiments.

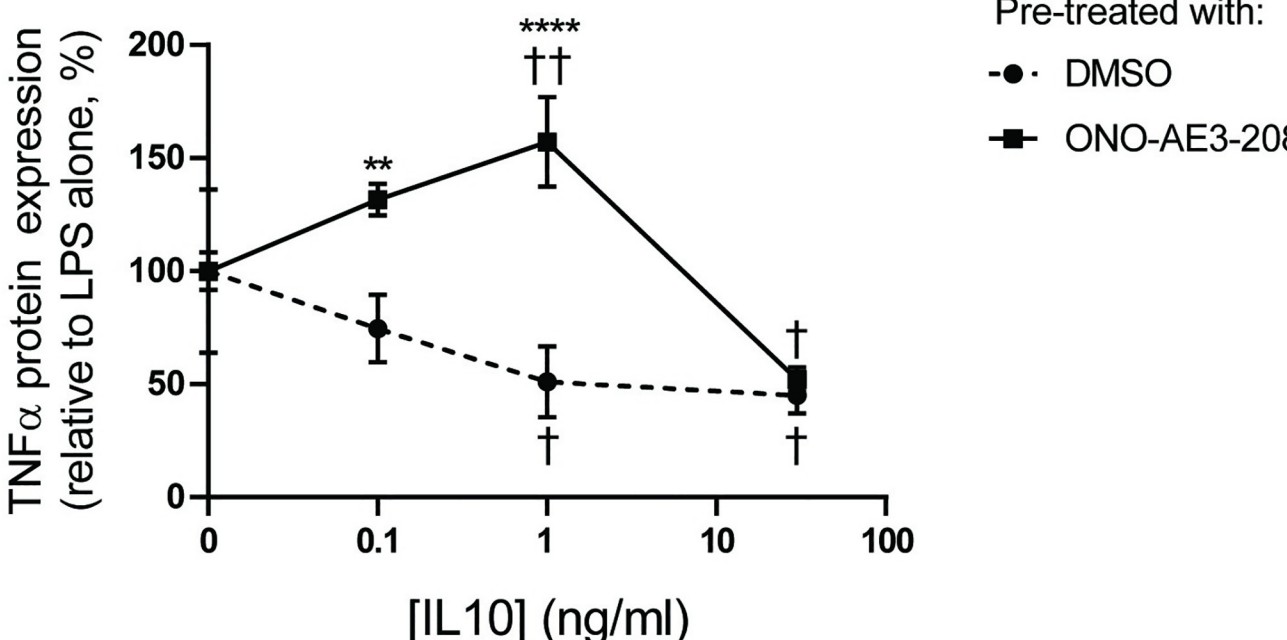

**Fig 5. EP4 antagonist ONO-AE3-208 prevents IL10 inhibition of TNFα.** RAW264.7 cells were pre-treated with 0.05% DMSO or with 100 nM ONO-AE3-208 for 1 hour prior to stimulation. RAW264.7 cells were stimulated with 1 ng/ml LPS +/- 0.1, 1.0, 30 ng/ml IL10 for 1 hour. Supernatants were collected to measure the level of secreted TNFα. The significance of the comparison between DMSO treatment and ONO-AE3-208 treatment was calculated by Two-Way ANOVA with Tukey's correction, **** $p < 0.0001$, *** $p < 0.001$, ** $p < 0.01$ * $p < 0.05$. The significance of the comparison between LPS and LPS + IL10 treatments was calculated by Two-Way ANOVA with Tukey's correction, †††† $p < 0.0001$, ††† $p < 0.001$, †† $p < 0.01$, † $p < 0.05$. Data are representative of three independent experiments.

and 6B). The IL10-dependent induction of CREB was abolished when EP4 protein is knocked down suggesting EP4 is required for IL10 induced phosphorylation of CREB (Fig 6A and 6B). Since elevation of cAMP levels has been shown to inhibit the activity of LPS-induced, PI3K-dependent signalling [49], we also examined whether IL10 inhibition of PI3K/Akt/mTOR pathway [46] required EP4 (Fig 6A and 6B). We found that IL10 inhibited LPS-activation of p85 subunit of PI3K but this inhibition was abolished when EP4 was knocked down (Fig 6A and 6B). Similarly, IL10 inhibition of the phosphorylation of Akt/mTOR target p70 S6K disrupted with the knock-down of EP4 suggesting that EP4 contributes to IL10 inhibition of PI3K/Akt/mTOR signalling and downstream processes.

Finally, we tested whether the knockdown of EP4 impacted IL10's ability to inhibit production of TNFα following LPS stimulation. We stimulated EP4 expressing (no Dox) or not-expressing (with Dox) RAW264.7 KD1/KD2 cells with LPS and various concentrations of IL10 to determine IC50 value for IL10 inhibition of TNFα production. We found for both EP4 KD1 and KD2 cells that knockdown of EP4 impaired IL10 inhibition of TNFα production as shown by IC50 values of 1.09 ± 0.2 ng/ml IL10 and 1.56 ± 0.62 ng/ml IL10 for Dox induced EP4 KD1 and KD2 versus 0.24 ± 0.01 ng/ml IL10 and 0.21 ± 0.03 ng/ml IL10 for the respective no Dox, EP4 expressing cells (Fig 6C).

## Discussion

EP4 levels have been reported to be upregulated in LPS activated macrophages [30]. We now show that elevation of EP4 *protein* (Figs 2–4) requires the presence of IL10, since LPS

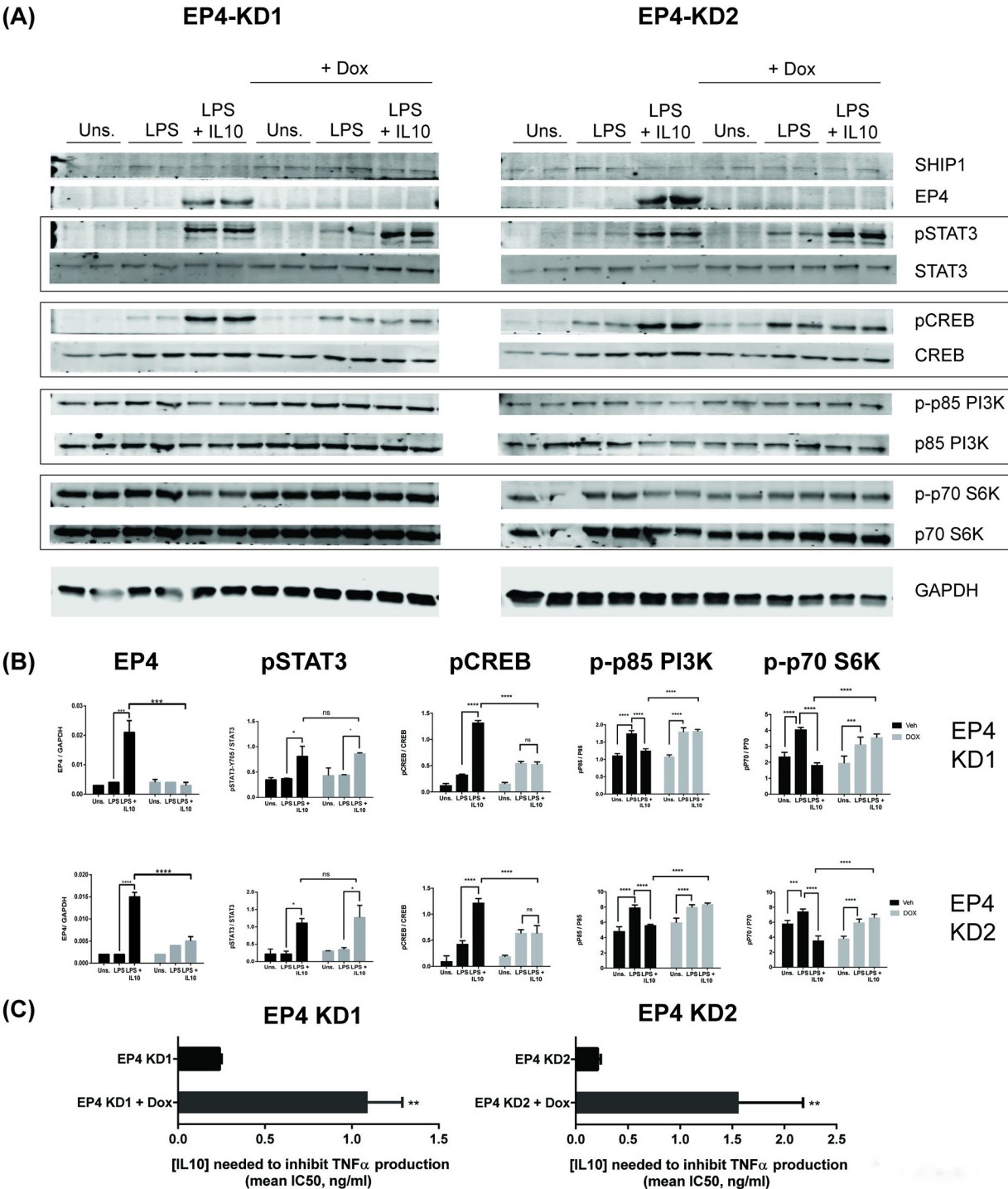

**Fig 6. EP4 is required for IL10 induction of phospho-CREB and inhibition of LPS-stimulation of phospho-p85 and phospho-p70 S6 kinase.**
RAW264.7/Cas9 cells transduced with EP4 sgRNA KD1 or KD2 gRNA were treated ± 2 μg/ml doxycycline for 48 hours to induce EP4 knockdown followed by stimulation with 10 ng/ml LPS ± 10 ng/ml IL10 for 1 hour and collection of protein lysates. Expression levels of EP4 and the indicated proteins and phospho-proteins were determined by immunoblotting (A), and quantifications shown in (B). (Two-Way ANOVA with Tukey's correction, **** $p < 0.0001$, *** $p < 0.001$, ** $p < 0.01$, * $p < 0.05$). Data are representative of three independent experiments. (C) The mean IC50 for IL10 inhibition of LPS-stimulated TNF$\alpha$ production. The significance of the comparison between no Dox and Dox treated EP4 KD cells was calculated by unpaired student's t-test, ** $p < 0.01$.

stimulation of macrophages for sufficient length of time results in production of IL10 [50, 51], but previous studies were not designed to separate the direct effects of LPS signaling through the TLR4 receptor from the subsequent actions of LPS-induced autocrine cytokines such as IL10 [52]. With respect to EP4 *mRNA* (Fig 1) we found LPS alone could increase levels in a SHIP1 and STAT3 independent manner in both perimacs and BMDM. The addition of IL10 + LPS increased EP4 mRNA levels much more than LPS alone, and appears to be SHIP1 dependent in both perimacs and BMDM. However, while IL10 induction of EP4 mRNA in perimacs is STAT3 dependent, in BMDM the induction is STAT3 independent. Future studies will examine whether IL10 dependent control of EP4 mRNA occurs transcriptionally or post-transcriptionally, as well as why EP4 mRNA responses with respect to STAT3 are different in BMDM compared to perimacs. IL10R signaling involves both SHIP1 and STAT3, and we found that IL10 upregulation of EP4 protein required both.

EP4 has been reported to mediate PGE$_2$'s inhibition of inflammatory mediators [31, 34]. Since LPS treatment of macrophages results in COX-2-dependent production of PGE$_2$ [53, 54], autocrine PGE$_2$ along with IL10-induction of EP4 protein may be a mechanism to self-limit the inflammatory response. Fig 7 shows our proposed model for the coordination and interplay between IL10 and PGE$_2$ signalling in controlling macrophage inflammation. LPS signalling (blue) leads to production of inflammatory mediators such as TNFα [8], but also of PGE$_2$ [53, 54] which binds to EP4 to inhibit inflammation [31, 36]. IL10 signalling (black)

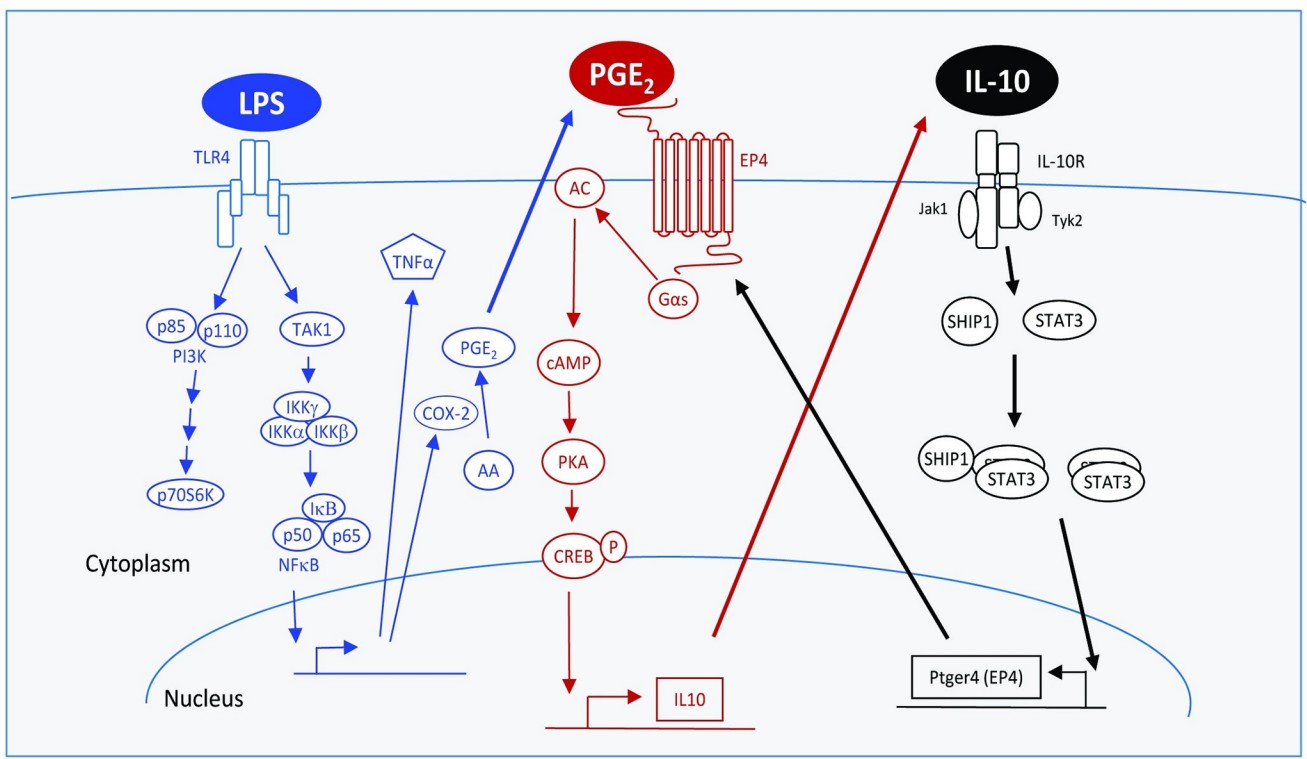

**Fig 7. IL10 and PGE2 signaling coordinate to inhibit macrophage inflammatory responses.** LPS signaling (blue) leads to production of inflammatory mediators such as TNFα, but also of PGE$_2$ which will have a net anti-inflammatory effect. IL10 signaling (black) leads to inhibition of LPS-induced inflammatory events partly through expression of genes which include EP4. EP4 signaling (red) involves the G$_{αs}$-dependent elevation of cAMP levels and subsequent activation and phosphorylation of the CREB transcription factor. pCREB interferes with LPS-induced NFκB transcription, and supports transcription of more IL10. Solid lines indicate protein-protein interactions; dashed lines indicate the production or action of ligands/proteins/second messengers. Abbreviations defined in the text.

leads to inhibition of LPS-induced inflammatory events partly through expression of genes which include EP4. EP4 signalling (red) involves G$_{\alpha s}$ activation of AC, elevation of cAMP levels and subsequent activation and phosphorylation of the CREB transcription factor [33]. pCREB interferes with LPS-induced NFκB transcription, as well as supports transcription of more IL10 [55]. Evidence for this model is discussed below.

Investigators have looked at whether PGE$_2$ stimulation of EP4 receptor dependent production of endogenous IL10 accounts for some or all of the anti-inflammatory action of PGE$_2$. Seldon *et al* used neutralizing anti-IL10 antibodies to show that PGE$_2$ inhibition of TNFα expression in human monocytes was independent of endogenous IL10 [56]. In contrast, studies with mouse perimacs cells and anti-IL10 antibodies [57], or mouse IL10$^{+/+}$ and IL10$^{-/-}$ BMDM [58] suggested that PGE$_2$ induction of TNFα or COX-2 depended on PGE$_2$ induction of IL10 expression. The apparent difference in IL10 requirement between the human and mouse studies might be due to differences in the kinetics of LPS-induced IL10 expression in the different cell types or their phenotype/differentiation state [57]. Regardless of whether EP4 agonists elicit all their anti-inflammatory actions through induction of endogenous IL10, an EP4 agonist has now been tested and shown to be effective in phase 2 trials of ulcerative colitis [41].

In this paper, we examined the conversed question of whether IL10R signalling involves subsequent PGE$_2$ stimulation of EP4 signalling. Corraliza *et al* [59] had found that IL10 treatment of BMDM activated PKA activity which peaked at 2–4 hours after IL10 addition. The delayed timing of PKA activation suggests that IL10R signalling indirectly leads to PKA activation, perhaps through EP4 signalling. In support of this, investigators have found that endogenous PGE$_2$ is synthesized in within an hour of LPS stimulation [60]). We show in this study that EP4 protein expression requires is expressed within an hour of IL10 addition (Fig 3) and that IL10 stimulation of pCREB required the presence of EP4 (Fig 6). Since elevation of cAMP levels has been shown to inhibit the activity of LPS-induced, PI3K-dependent signalling [49] in macrophages, we examined whether IL10 inhibition of PI3K signalling required EP4. We found that IL10 inhibition of LPS-stimulated phosphorylation of p85 subunit of PI3K and the downstream p70 S6 kinase does require the presence of EP4 protein (Fig 6).

Altogether, these data are consistent with our model where EP4 signalling might be responsible for the activation of PKA-dependent signalling previously ascribed to the immediate signalling from the IL10R. That IL10 activation of cAMP signalling occurs indirectly through EP4 also provides an explanation for why cAMP elevating agents are synergistic rather than additive with IL10 stimulation for induction of certain inflammatory proteins [61]. Most importantly, since PGE$_2$ agonists exert their anti-inflammatory action through EP4, only patients whose inflammatory cells express EP4 will benefit from treatment. We show in this study that IL10 can induce EP4 protein expression; it remains to be determined whether other factors can lead to EP4 expression in normal physiology and inflammatory disease. If IL10 is the main or only cytokine which upregulates EP4 protein expression in macrophages then PGE$_2$ agonists may not be effective in diseases in which the IL10R or IL10 is deficient [12, 17, 62].

## Supporting information

**S1 Raw images.**
(PDF)

## Acknowledgments

We thank Dr. Gerald Krystal for the SHIP1$^{-/-}$ mice and Dr. Shizuo Akira for the STAT3$^{fl/fl}$ mice.

## Author Contributions

**Conceptualization:** Sylvia T. Cheung, Alice L. -F. Mui.

**Data curation:** Abrar Samiea, Jeff S. J. Yoon, Thomas C. Chamberlain, Alice L. -F. Mui.

**Formal analysis:** Abrar Samiea, Thomas C. Chamberlain, Alice L. -F. Mui.

**Funding acquisition:** Alice L. -F. Mui.

**Investigation:** Abrar Samiea, Jeff S. J. Yoon, Sylvia T. Cheung, Thomas C. Chamberlain, Alice L. -F. Mui.

**Methodology:** Abrar Samiea, Jeff S. J. Yoon, Sylvia T. Cheung, Thomas C. Chamberlain, Alice L. -F. Mui.

**Project administration:** Abrar Samiea, Jeff S. J. Yoon, Sylvia T. Cheung, Thomas C. Chamberlain, Alice L. -F. Mui.

**Resources:** Jeff S. J. Yoon, Sylvia T. Cheung, Thomas C. Chamberlain, Alice L. -F. Mui.

**Supervision:** Sylvia T. Cheung, Thomas C. Chamberlain, Alice L. -F. Mui.

**Validation:** Abrar Samiea, Thomas C. Chamberlain, Alice L. -F. Mui.

**Visualization:** Abrar Samiea, Thomas C. Chamberlain, Alice L. -F. Mui.

**Writing – original draft:** Abrar Samiea, Thomas C. Chamberlain, Alice L. -F. Mui.

**Writing – review & editing:** Jeff S. J. Yoon, Sylvia T. Cheung, Thomas C. Chamberlain, Alice L. -F. Mui.

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
