## [Decision Letter · Decision Letter 0]

18 Nov 2019

PONE-D-19-28373

Interleukin-10 contributes to PGE2 signalling through upregulation of EP4 via SHIP1 and STAT3

PLOS ONE

Dear Dr. Mui,

Thank you for submitting your manuscript to PLOS ONE. After careful consideration, we feel that it has merit but does not fully meet PLOS ONE’s publication criteria as it currently stands. Therefore, we invite you to submit a revised version of the manuscript that addresses the points raised during the review process.

In particular, Rev 1 made several critical comments that have to be addressed before the manuscript can be reconsidered for publication including consistency of experimental systems (this requires at least additional controls and potential revision of conclusions), use of adequate controls for determining effects of Il10 on EP4 and role of LPS in that phenomenon, and, proper normalization of western blots as well as validation of nuclear localization-perturbing constructs. 

We would appreciate receiving your revised manuscript in 90 days from the date of this decision letter. To enhance the reproducibility of your results, we recommend that if applicable you deposit your laboratory protocols in protocols.io, where a protocol can be assigned its own identifier (DOI) such that it can be cited independently in the future. For instructions see: http://journals.plos.org/plosone/s/submission-guidelines#loc-laboratory-protocols

We look forward to receiving your revised manuscript.

Kind regards,

Michal Hetman

Academic Editor

PLOS ONE

Journal Requirements:

3. Please include a caption for figure 7.

Reviewers' comments:

Reviewer's Responses to Questions

**Comments to the Author**

1. Is the manuscript technically sound, and do the data support the conclusions?

Reviewer #1: Partly

Reviewer #2: Yes

2. Has the statistical analysis been performed appropriately and rigorously? 

Reviewer #1: Yes

Reviewer #2: Yes

3. Have the authors made all data underlying the findings in their manuscript fully available?

Reviewer #1: Yes

Reviewer #2: Yes

4. Is the manuscript presented in an intelligible fashion and written in standard English?

Reviewer #1: Yes

Reviewer #2: Yes

5. Review Comments to the Author

Reviewer #1: The manuscript by Samiea et al describes a potential mechanism by which the anti-inflammatory activities of IL-10 are manifested in macrophages. The authors find that IL-10 induces upregulation of the prostaglandin E2 receptor EP4 and this requires both STAT3 and SHIP1. Using a variety of knockdown systems, the authors present data suggesting that IL-10-mediated upregulation of EP4 and subsequent EP4 signaling is required to elicit the anti-inflammatory activities of IL-10. While the manuscript is presented in a clear and logical fashion, the quality of the some of the figures makes the data difficult to interpret. Furthermore, there are important controls missing from some of the figures.

Specific comments.

1. The data presented in figure 1 suggests that IL-10 upregulates EP4 mRNA in peritoneal macrophages in the presence of LPS, and this is dependent on both SHIP1 and STAT3. The data in figure 2 suggests that IL-10 induces EP4 protein in a macrophage cell line and a bone marrow-derived macrophage line derived from SHIP1 knockout mice that has been reconstituted with SHIP1, and this is independent of LPS. Is the upregulation of EP4 mRNA in figure 1 also independent of LPS? The use of the different cell lines in the different figures makes the data difficult to interpret since IL-10 can have differential effects on different cell types. The authors should be consistent in the systems they use to test theit hypotheses.

2. Since it appears that LPS is not required for induction of IL-10-mediated EP4 protein induction, effects on (figure 2), the authors should consider including looking at the effects of IL-10 alone in their efforts to elucidate the IL-10/EP4 signaling axis.

3. How are the authors quantitating STAT3 activation in the STAT3 knockout cells in figure 4C? There is no STAT3 expressed in those cells.

4. The authors should show that the different point mutants of SHIP1 used for the data in figure 5 are indeed defective for nuclear translocation.

5. From figure 5B, it appears that LPS and LPS + IL-10 decrease the expression of the K331A mutant SHIP1. This needs to be clarified. Is it possible this decrease in the K331A SHIP1 level is sufficient to impair the induction of EP4?

6. The total levels of STAT3 are missing from figure 5 which should be used to normalize activation of STAT3, rather than Actin. Similarly, the levels of CREB, p85-PI3K and p70 S6K are required for normalization of the concomitant phosphor proteins in figure 6. The data is difficult to interpret without these important controls.

7. The data presented in figure 6 suggests that EP4 is required for IL-10 activity. The use of an alternate method to verify this would strengthen the manuscript. Macrophages from EP4 knockout mice or an EP4 antagonist could be used to confirm these data.

8. The western blots for EP4 in figures 3 and 4A are not clear, making it difficult to interpret the data.

9. There is no point to including an Actin loading control in Figure 4 if it is not used as a normalizer for quantitation.

Reviewer #2: Interesting study well supported by the experimental data, adding to the existing information on PGE2 and IL-10 Rs. The authors wrote aclear report on their data which add to the existing information on IL-10 signaling including now the upregulation of EP4 in macrophages

6. PLOS authors have the option to publish the peer review history of their article (what does this mean?). If published, this will include your full peer review and any attached files.

Reviewer #1: No

Reviewer #2: No

---

## [Author Response · Author response to Decision Letter 0]

7 Feb 2020

Reviewer #1 is the only one requiring changes. We thank the reviewer for their suggestions, and have added new data as requested (points #1, 6 and 7 below). We also removed all data related to the J17 cells reconstituted with SHIP1 mutants (points #4 and 5). Corrections were also made as suggested.

As requested by the Editor, we have provided original uncropped and unadjusted images underlying all blot results reported in our figures. These images can be found within the file S1_raw_images.pdf within Supporting Information.

Reviewer #1: The manuscript by Samiea et al describes a potential mechanism by which the anti-inflammatory activities of IL-10 are manifested in macrophages. The authors find that IL-10 induces upregulation of the prostaglandin E2 receptor EP4 and this requires both STAT3 and SHIP1. Using a variety of knockdown systems, the authors present data suggesting that IL-10-mediated upregulation of EP4 and subsequent EP4 signaling is required to elicit the anti-inflammatory activities of IL-10. While the manuscript is presented in a clear and logical fashion, the quality of the some of the figures makes the data difficult to interpret. Furthermore, there are important controls missing from some of the figures. Specific comments.

1. The data presented in figure 1 suggests that IL-10 upregulates EP4 mRNA in peritoneal macrophages in the presence of LPS, and this is dependent on both SHIP1 and STAT3. The data in figure 2 suggests that IL-10 induces EP4 protein in a macrophage cell line and a bone marrow-derived macrophage line derived from SHIP1 knockout mice that has been reconstituted with SHIP1, and this is independent of LPS. Is the upregulation of EP4 mRNA in figure 1 also independent of LPS? The use of the different cell lines in the different figures makes the data difficult to interpret since IL-10 can have differential effects on different cell types. The authors should be consistent in the systems they use to test their hypotheses.

ANSWER: As suggested by the reviewer, we have now looked at EP4 mRNA levels in each of the cell types (perimacs, BMDM and RAW264.7 cells) used in our study. We chose only to look at the action of IL10 in the presence of LPS activation, since the physiological target of IL10 is the activated macrophage (1). The new mRNA expression data from bone marrow derived macrophages (BMDM, Fig 1B and 1C) and RAW264.7 cells (Fig 1D) have been added to Fig 1. 

We found that EP4 mRNA induction by IL10 required SHIP1 in both perimacs (Fig 1A, left) and BMDM (Fig 1B, left), since the level of EP4 mRNA induced by IL10 is significantly diminished in SHIP1-/- perimacs and BMDM. However, although STAT3 seems to contribute to IL10 induction of EP4 in perimacs (Fig 1A, right), in BMDM, STAT3 is not required for EP4 induction (Fig 1B, right). We checked that the STAT3-/- are impaired in IL10 responses as we and others have shown (2, 3), and confirmed IL10 could not inhibit TNFα expression in STAT3-/- BMDM (Fig 1C). 

The focus of the current paper is the IL10 regulation of EP4 protein expression. In future studies we will examine whether the increased mRNA occurs transcriptionally or post-transcriptionally, whether IL10 alone is able to elicit these changes, and why the STAT3 dependence of IL10 induction of EP4 mRNA occurs only in perimacs and not BMDM.

2. Since it appears that LPS is not required for induction of IL-10-mediated EP4 protein induction, effects on (figure 2), the authors should consider including looking at the effects of IL-10 alone in their efforts to elucidate the IL-10/EP4 signaling axis.

ANSWER: The physiological target of IL10 is the activated macrophage (1). IL10 is produced by an activated macrophage as an autocrine, negative feedback inhibitor. IL10 inhibits the macrophage production of TNFα (Fig 5, Fig 6C), and LPS induced signaling events (Fig 6 A/B). Furthermore, the consequence of EP4 expression (induced by IL10) is only seen in if the EP4 ligand (PGE2) is present. PGE2 is induced by LPS. We have removed the data of the studies in which IL10 is added to inactivated macrophages (old Fig 2).

3. How are the authors quantitating STAT3 activation in the STAT3 knockout cells in figure 4C? There is no STAT3 expressed in those cells.

ANSWER: We apologize, this was an error. The pSTAT3 and STAT3 quantification in STAT3 knockout cells has been removed.

4. The authors should show that the different point mutants of SHIP1 used for the data in figure 5 are indeed defective for nuclear translocation.

We are unable to redo the experiments in the cells reconstituted with different SHIP1 mutants since these cells have been lost (LN2 Dewar thawed). 

ANSWER: We have removed all the data related to analysis of effect of SHIP1 mutations on IL10 induction of EP4 protein. These analyses of the contribution of SHIP1’s nuclear localization signal is peripheral to our main message that IL10 induction of EP4 protein is SHIP1 dependent and contributes to IL10 inhibition of macrophage activation as shown by TNFα production.

5. From figure 5B, it appears that LPS and LPS + IL-10 decrease the expression of the K331A mutant SHIP1. This needs to be clarified. Is it possible this decrease in the K331A SHIP1 level is sufficient to impair the induction of EP4?

ANSWER: Yes, it is possible the diminished expression of K331A rather than the K331A substitution is responsible for the impaired EP4 induction. However, we have removed all the data related to the SHIP1 mutants from the paper.

6. The total levels of STAT3 are missing from figure 5 which should be used to normalize activation of STAT3, rather than Actin. Similarly, the levels of CREB, p85-PI3K and p70 S6K are required for normalization of the concomitant phosphor proteins in figure 6. The data is difficult to interpret without these important controls.

ANSWER: The levels of protein STAT3 CREB, p85-PI3K and p70-S6K were measured and used to normalize the level of phosphorylated proteins (Revised Fig 6). 

7. The data presented in figure 6 suggests that EP4 is required for IL-10 activity. The use of an alternate method to verify this would strengthen the manuscript. Macrophages from EP4 knockout mice or an EP4 antagonist could be used to confirm these data.

ANSWER: Thank-you for suggesting the use of an EP4 antagonist. When the EP4 antagonist, RAW264.7 cells are treated with 100 nM ONO-AE3-208 IL-10 was no longer able to inhibit TNFα production (New Fig 5); rather the level of TNFα actually increased in cells treated with low (0.1 – 1 ng/mL) concentration of IL10 concentrations. High (30 ng/mL) concentrations of IL10 was able to overcome the effect of ONO-AE3-208. 

8. The western blots for EP4 in figures 3 and 4A are not clear, making it difficult to interpret the data.

ANSWER: We have replaced the EP4 blots in Fig 3 and 4A.

9. There is no point to including an Actin loading control in Figure 4 if it is not used as a normalizer for quantitation.

ANSWER: We have removed Actin loading control was removed from the Fig 4.

References:

1. Iyer SS, Cheng G. Role of interleukin 10 transcriptional regulation in inflammation and autoimmune disease. Crit Rev Immunol. 2012;32(1):23-63.

2. Chan CS, Ming-Lum A, Golds GB, Lee SJ, Anderson RJ, Mui AL-F. Interleukin-10 inhibits LPS induced TNFα translation through a SHIP1-dependent pathway. Journal of Biological Chemistry. 2012.

3. Cheung ST, So EY, Chang D, Ming-Lum A, Mui AL. Interleukin-10 inhibits lipopolysaccharide induced miR-155 precursor stability and maturation. PLoS One. 2013;8(8):e71336.

---

## [Decision Letter · Decision Letter 1]

2 Mar 2020

Interleukin-10 contributes to PGE2 signalling through upregulation of EP4 via SHIP1 and STAT3

PONE-D-19-28373R1

Dear Dr. Mui,

We are pleased to inform you that your manuscript has been judged scientifically suitable for publication and will be formally accepted for publication once it complies with all outstanding technical requirements.

With kind regards,

Michal Hetman

Academic Editor

PLOS ONE

Additional Editor Comments (optional):

Reviewers' comments:

Reviewer's Responses to Questions

**Comments to the Author**

1. If the authors have adequately addressed your comments raised in a previous round of review and you feel that this manuscript is now acceptable for publication, you may indicate that here to bypass the “Comments to the Author” section, enter your conflict of interest statement in the “Confidential to Editor” section, and submit your "Accept" recommendation.

Reviewer #1: All comments have been addressed

2. Is the manuscript technically sound, and do the data support the conclusions?

Reviewer #1: Yes

3. Has the statistical analysis been performed appropriately and rigorously? 

Reviewer #1: Yes

4. Have the authors made all data underlying the findings in their manuscript fully available?

Reviewer #1: Yes

5. Is the manuscript presented in an intelligible fashion and written in standard English?

Reviewer #1: Yes

6. Review Comments to the Author

Reviewer #1: (No Response)

7. PLOS authors have the option to publish the peer review history of their article (what does this mean?). If published, this will include your full peer review and any attached files.

Reviewer #1: No

---

## [Editor Report · Acceptance letter]

6 Mar 2020

PONE-D-19-28373R1 

Interleukin-10 contributes to PGE2 signalling through upregulation of EP4 via SHIP1 and STAT3 

Dear Dr. Mui:

I am pleased to inform you that your manuscript has been deemed suitable for publication in PLOS ONE. Congratulations! Your manuscript is now with our production department. 

With kind regards,

on behalf of

Dr. Michal Hetman 

Academic Editor

PLOS ONE